# FIBER MONTE CARLO

**Nick Richardson**
Department of Computer Science
Princeton University
Princeton, NJ 08544
njkrichardson@princeton.edu

**Deniz Oktay**
Department of Computer Science
Princeton University
Princeton, NJ 08544
doktay@princeton.edu

**Yaniv Ovadia**
Department of Computer Science
Princeton University
Princeton, NJ 08544
ovadia@princeton.edu

**James C. Bowden**
EECS Department
UC Berkeley*
Berkeley, CA 94720
jcbowden@berkeley.edu

**Ryan P. Adams**
Department of Computer Science
Princeton University
Princeton, NJ 08544
rpa@princeton.edu

## ABSTRACT

Integrals with discontinuous integrands are ubiquitous, arising from discrete structure in applications like topology optimization, graphics, and computational geometry. These integrals are often part of a forward model in an inverse problem where it is necessary to reason backwards about the parameters, ideally using gradient-based optimization. Monte Carlo methods are widely used to estimate the value of integrals, but this results in a non-differentiable approximation that is amenable to neither conventional automatic differentiation nor reparameterization-based gradient methods. This significantly disrupts efforts to integrate machine learning methods in areas that exhibit these discontinuities: physical simulation and robotics, design, graphics, and computational geometry. Although bespoke domain-specific techniques can handle special cases, a general methodology to wield automatic differentiation in these discrete contexts is wanting. We introduce a differentiable variant of the simple Monte Carlo estimator which samples line segments rather than points from the domain. We justify our estimator analytically as conditional Monte Carlo and demonstrate the diverse functionality of the method as applied to image stylization, topology optimization, and computational geometry.

## 1 INTRODUCTION

Determining the value of an integral is a problem at the core of myriad applications across machine learning, statistics, and computational science and engineering. For example, in graphics, pixel shaders in rasterizing rendering engines compute a spatial integral over a collection of geometric primitives to compute an image. In machine learning and statistical inference, integrals abound in operationalizing probabilistic inference systems: marginalization and normalization are both fundamental to Bayesian computation.

Modeling the generative processes associated with these applications involves computing (or at minimum, estimating) the value of these integrals in the forward model. In applications with discrete structure, the forward model often contains integrals with parametric discontinuities. That is to say, integrals whose integrand contains some discontinuous expression (with respect to a parameter of

---

*Work done while an undergraduate at Caltech.

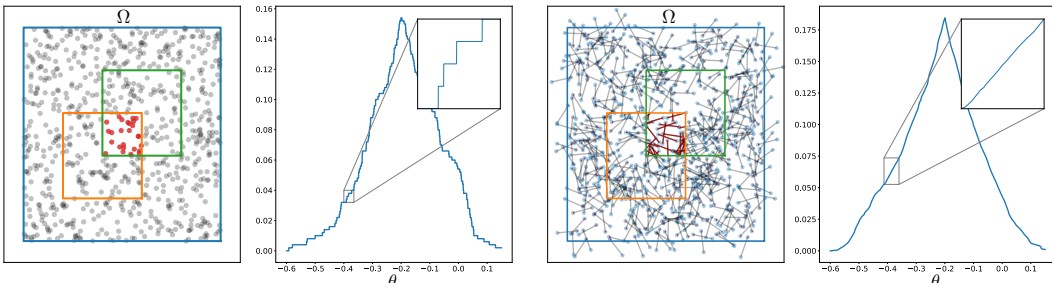

Figure 1: Left: the simple Monte Carlo estimator for intersection over union increases/decreases value on the order $O(1/n)$ as the green, moveable shape $P_2$ is translated across the fixed orange shape $P_1$ along $(-\theta, -\theta)^T$, even in the limit of a zero magnitude translation amount. This is because dropping points with zero extent means the number of points contained in the intersection (each of which increases/decreases the estimate by $O(1/n)$), increases/decreases at discrete points in the domain of $\theta$. Right: Fiber Monte Carlo results in a piecewise linear relationship between the estimated intersection over union and the amount of translation.

the model). As a simple concrete case, consider two polygons $P_1, P_2 \in \Omega \subseteq \mathbb{R}^2$; squares, for instance. Suppose the location of $P_1$ is fixed, but we can choose an amount $\theta \in \mathbb{R}$ to translate $P_2$ along the vector $(-1, -1)^T$ to maximize the area of intersection $\mathcal{L} : \mathbb{R} \mapsto \mathbb{R}$ of the shapes (see fig. 1). This corresponds with an optimization problem whose objective is an integral with parametric discontinuity (with respect to $\theta$):

$$\mathcal{L}(\theta) = \int_\Omega \mathbb{I}[\mathbf{x} \in (P_1 \cap P_2(\theta))]d\mathbf{x}. \tag{1}$$

Intuitively, integrals with parametric discontinuities like this arise from, e.g., hard spatial boundaries and collisions between objects in simulating physics, shadowing and overlap phenomena in a rendering pipeline, or the discrete geometry in topology optimization. With the increasing popularity of machine learning systems that are tightly coupled with physical simulation, rendering, and other geometric computation, these integrals are ubiquitous. We could use simple Monte Carlo to estimate the value of an integral with a parametric discontinuity, but we want to reason backwards about the parameters (using the derivatives $\nabla_\theta \mathcal{L}$) and simple Monte Carlo results in a nondifferentiable estimate (see fig. 1), even though $\mathcal{L}$ is differentiable in exact form.

The prospect of automatically differentiating these programs with respect to the parameters is enticing, but conventional automatic differentiation systems do not compute derivatives of these parametric discontinuities correctly, so general gradient-based optimization techniques cannot be applied directly (Bangaru et al., 2021). Existing optimization methods in these contexts involve bespoke solutions: smoothing of the discontinuous integrands (Liu et al., 2019), ignoring discontinuities (Gkioulekas et al., 2013), including discontinuities as constraints and (locally) solving nonlinear programs (Li et al., 2020), and more recently domain-specific differentiation languages (Bangaru et al., 2021). To our knowledge, no existing methodology can utilize generic automatic differentiation frameworks, which offer the advantage of domain-independent computation of derivative functions, higher-order differentiation, and mapping computations to accelerator backends (Jouppi et al., 2017) via optimizing compilers (Frostig et al., 2018).

In this work, we introduce **Fiber Monte Carlo**, a variant of simple Monte Carlo which enables differentiable integral estimates, even when the integrand contains a parametric discontinuity. Fiber Monte Carlo samples line segments from the domain rather than points, and adds up the continuous *amount* of those line segments under a function rather than the discrete *number* of points under a function. For low-dimensional, physics/geometry-oriented problems like graphics, topology optimization, computational geometry, and physical simulation, Fiber Monte Carlo can be employed to estimate these problematic integrals in an unbiased and differentiable way: enabling gradient-based optimization. It is important to note that Fiber Monte Carlo is appropriate provided the *domain of integration* has small dimension. This is unrelated to the dimension of the parameter: we show that there is no difficulty in training large parametric models (e.g., neural networks) with this framework. Further, we implement a variety of generic Fiber Monte Carlo estimators in a general-purpose automatic differentiation library, JAX (Bradbury et al., 2018).

We further illustrate the ubiquity of integrals with parametric discontinuities and the generality of the estimator by utilizing Fiber Monte Carlo in applications spanning graphics, topology optimization, and computational geometry. In section 4.1 we describe an image stylization application in which we build a simple differentiable rendering engine, using Fiber Monte Carlo as a method for implementing probabilistic pixel shaders with neural field rendering primitives. In section 4.2 we demonstrate a differentiable method for topology optimization of mechanical structures. In section 4.3 we describe a method for the amortized computation of approximate convex hulls using neural networks, with applications to membership oracles and intersection primitives that inform a variety of problems in computational geometry.

## 2 RELATED WORK

### 2.1 MONTE CARLO GRADIENT ESTIMATION

Methods to compute derivatives of the expectation of a function (so-called stochastic derivatives) are widely studied (Mohamed et al., 2020). These methods are concerned with a class of optimization problems in which the objective $\mathcal{L}$ can be written as the expectation of a random variable $x$ under a 'cost function' $f$, with respect to a distribution $p$ (with parameter $\theta$). Often the cost function is also parametric (with parameter $\phi$), but in the standard formulation the optimization variable is the parameter $\theta$ of the distribution $p$. The objective is often written as

$$\mathcal{L}(\theta) = \mathbb{E}_{p_\theta(x)}[f_\phi(x)] = \int_{\text{dom } x} p_\theta(x) f_\phi(x) dx.$$

To use a gradient method, one aims to estimate derivatives $\nabla_\theta \mathcal{L}(\theta)$ of the objective. At a coarse level, the taxonomy of gradient estimators contains two groups. The first compute derivatives by differentiation of the measure $p_\theta(x)$. This class includes the score function estimator (Glynn, 1990; Williams, 1992) and the measure-valued gradient (Pflug, 2012). In relation to Fiber Monte Carlo, these estimators can similarly be employed against discontinuous (even black-box) cost functions $f$, but differ in that they do not use derivative information from the cost function $f$.

The second class of gradient estimators compute derivatives of "paths", i.e., cleverly designed additions to a computation graph which elicit a dependency of the cost $f_\phi(x)$ through the distributional parameters $\theta$. Pathwise gradient estimators are popular in machine learning under the name "reparameterization" (Kingma & Welling, 2013) and "stochastic backpropagation" (Rezende et al., 2014), and appear in perturbation analysis (Glasserman & Ho, 1991), financial engineering (Glasserman, 2004), and optimization (Pflug, 2012) dating back decades. Pathwise gradient estimators are similar to Fiber Monte Carlo in that they exploit structural characteristics of the cost $f$ and are thus less general, but differ in that pathwise derivative methods require that $f$ is differentiable with respect to the optimization variable. Fiber Monte Carlo is applicable in low-dimensional settings and is likewise less general as compared to methods using derivatives of measure, but in contrast to path methods, it obviates the necessity of a differentiable cost $f$,

### 2.2 DIFFERENTIATING PARAMETRIC DISCONTINUITIES

Intriguing recent work considers the project of differentiating integrals with parametric discontinuities (Bangaru et al., 2021; Yang et al., 2022; Morozov et al., 2023). These works introduce domain specific languages for computing these derivatives, aimed primarily at applications in graphics and differentiable rendering. As we see it, the core contributions are domain-specific languages which, unlike conventional automatic differentiation libraries, correctly compute the derivatives in question, manipulating the Dirac delta contributions from the discontinuities directly and sometimes even including integration as a language primitive (Bangaru et al., 2021). That said, the languages are constrained in their ability to flexibly and automatically manipulate the computation graph, resulting in more burden on the programmer to formulate the application in ways amenable to the compiler (e.g., representing discontinuities via affine functions, ensuring that an integrand is diffeomorphic, etc.). For example, Bangaru et al. (2021) utilizes a source-to-source automatic differentiation framework (Griewank & Walther, 2008). The language cannot differentiate through control flow structures or implement runtime-determined looping (i.e., it is not Turing complete). This is no shortcoming in the context of graphics applications, but represents a limitation in its applicability and wider adoption.

## 3 METHOD

The probabilistic dual between expectations and integrals is actually instructive in understanding exactly what is problematic about computing derivatives in these contexts. Simple Monte Carlo estimates integrals using random objects with precisely zero spatial extent: points. Put another way, varying the parameters of a discontinuous function over the domain means the value of the estimate can change significantly (on the order of $1/n$) by suddenly including/excluding a point even with an arbitrarily small modulation of the parameter (see fig. 1).

### 3.1 SAMPLING METHOD

The sampling method utilizes a uniform distribution over line segments (fibers) in the domain of integration; sampling fibers uniformly is the condition which implies correct (unbiased) estimates. In section 3.3 we explain that the key to differentiating these estimates is the use of objects which have extent, and not only location.

Given a compact set, the 'sampling domain' $\Omega \subset \mathbb{R}^d$, the model is comprised of uniform distributions over (1) one 'start' point $\mathbf{x}_s$ of each fiber (sampled uniformly over $\Omega$) and (2) the surface of a norm ball with fixed radius $\ell > 0$ centered at $\mathbf{x}_s$. We associate with $\Omega$ an 'extended' domain $\overline{\Omega} = \{\mathbf{x} \mid \mathbf{dist}(\mathbf{x}, \Omega) \leq \ell\}$, which is the original domain and an extra 'shell' of width $\ell$ which forms the domain for the endpoint of each fiber[1]. The sampling procedure can be written:

$$\mathbf{x}_s \sim \text{Uniform}(\Omega), \quad \mathbf{z} \sim \mathcal{N}(0, \mathbb{I}_d). \tag{2}$$

Since the length $\ell$ is non-random, the endpoint $\mathbf{x}_e \in \overline{\Omega}$ of the fiber is determined as:

$$\mathbf{x}_e = \mathbf{z}\frac{\ell}{||\mathbf{z}||}. \tag{3}$$

The estimator induced using fibers as random objects is analogous to the simple Monte Carlo estimator. To estimate the population expectation of a function $h : \mathbb{R}^d \mapsto \mathbb{R}$ of a random variable, simple Monte Carlo forms the sample expectation of the value of $h$ over a collection of *points*. Instead, we form the sample expectation of the line integral of $h$ over a collection of fibers. The estimate, given a collection of $n$ fibers of length $\ell$ is:

$$\hat{\mu}_n = \frac{1}{n\ell} \sum_{i=1}^{n} \int_{t=0}^{t=1} h(r_i(t))|r_i'(t)|dt, \tag{4}$$

where $r_i(t) : [0, 1] \mapsto \mathbb{R}^d$ is a bijective parameterization of the fiber. Section 3.2 explains why this estimator is correct and produces correct derivatives.

At first blush, it appears that Fiber Monte Carlo applies only in contexts where one can somehow analytically describe the line integral of the function. Operationally, we use an implicit function formulation in conjunction with implicit differentiation to obviate this apparent requirement (see section 3.3). In our formulation, we need only pointwise function evaluations to evaluate the estimator (akin to simple Monte Carlo).

### 3.2 CORRECTNESS

In this section we explain why the Fiber Monte Carlo estimator is correct, and provide a a simple characterization of the variance. To show correctness, we must demonstrate that the induced probability distribution over points $x \in \Omega$ is uniform. We first the describe the condition on the distribution of fibers that is required to ensure the distribution is uniform, and then show that the sampling method described in Section 3.1 obeys this condition.

In fiber sampling, we first pick a set $F$ of equal length 1-D subsets (intervals) from the extended domain, $F \subset 2^{\overline{\Omega}}$, and a probability measure $p_F : F \to \mathbb{R}$ over them. We sample a subset $f \sim p_F$, and then sample a point $x \in f$ uniformly. The marginal probability $p(x)$ is given as:

$$p(x) = \int_{f \in F} p(x|f)dp_F(f). \tag{5}$$

---

[1]$\overline{\Omega}$ is strictly larger since start points near the boundary of $\Omega$ may induce endpoints outside of $\Omega$.

Since $x$ is sampled uniformly on $f$, $p(x|f) = \ell \, \mathbb{I}[x \in f]$, where $\ell = \mathbf{len} \, f$ is independent of $f$. We then have

$$p(x) = \ell \int_{f \in F} \mathbb{I}[x \in f] dp_F(f) = \ell \, p_F(x \in f), \tag{6}$$

which reduces the condition of uniform $p(x)$, $x \in \Omega$ to $p_F(x \in f) = c_1$, $\forall x \in \Omega$, for some constant $c_1$. Note that technically $p(x)$ is defined over the extended domain $\overline{\Omega}$, but we can use rejection sampling to get a distribution over $\Omega$. This condition can be summarized as: the probability of each $x$ being contained in a sampled fiber should be independent of $x$. The condition holds since, as we explain in section 3.1, the extended domain allows us to ensure uniformity on points close to the boundary.

In Fiber Monte Carlo, we wish to compute derivatives of expectations (or integrals) containing parametric discontinuities. In the following, we let $a \sim A$ for set $A$ denote $a \sim \mathrm{Unif}(A)$. We assume $h_\theta : \Omega \to \mathbb{R}$ is discontinuous in $\Omega$, but that $\mathbb{E}_{x \sim \Omega}[h_\theta(x)]$ is continuous and (sub-)differentiable in $\theta$. Furthermore, we assume $\mathbb{E}_{x \sim f}[h_\theta(x)|f]$, the expectation over each fiber, is continuous and (sub-)differentiable in $\theta$ for almost all $f \in F$.

As a result of the discontinuity,

$$\nabla_\theta \mathbb{E}_{x \sim \Omega}[h_\theta(x)] \neq \mathbb{E}_{x \sim \Omega}[\nabla_\theta h_\theta(x)] \tag{7}$$

so we cannot use the Monte Carlo estimate formed by point-wise gradient evaluations (which is what automatic differentiation frameworks will do). In Fiber Monte Carlo, we decompose the expectation into two parts, one of which is analytically computable. Per our results above,

$$\mathbb{E}_{x \sim \Omega}[h_\theta(x)] = \mathbb{E}_{f \sim F}\left[\mathbb{E}_{x \sim f}[h_\theta(x)|f]\right]. \tag{8}$$

Since by assumption $\mathbb{E}_{x \sim f}[h_\theta(x)|f]$ is almost everywhere (up to a set of zero measure) a continuous function of $\theta$, we can exchange expectation and differentiation as in

$$\nabla_\theta \mathbb{E}_{x \sim \Omega}[h_\theta(x)] = \nabla_\theta \mathbb{E}_{f \sim F}\left[\mathbb{E}_{x \sim f}[h_\theta(x)|f]]\right] \tag{9}$$
$$= \mathbb{E}_{f \sim F}\left[\nabla_\theta \mathbb{E}_{x \sim f}[h_\theta(x)|f]]\right], \tag{10}$$

using Leibniz' rule. We compute $\nabla_\theta \mathbb{E}_{x \sim f}[h_\theta(x)|f]$ exactly through the implicit function theorem, described in Section 3.3. Averaging over fibers gives an unbiased estimate of $\nabla_\theta \mathbb{E}_{x \sim \Omega}[h_\theta(x)]$.

We can show that this is a variant of conditional Monte Carlo (Owen, 2013). Let $h_\theta(x)$ denote a random variable with $x \sim \Omega$, and $E_{x \sim f}[h_\theta(x)|f]$ be a random variable with $f \sim F$. $\mathrm{Var}[h_\theta(x)]$ and $\mathrm{Var}[E_{x \sim f}[h_\theta(x)|f]]$ is the variance for simple Monte Carlo and Fiber Monte Carlo, respectively. By the Law of Total Variance,

$$\mathrm{Var}[h_\theta(x)] = \mathbb{E}_{f \sim F}\left[\mathrm{Var}_{x \sim f}[h_\theta(x)]\right] + \mathrm{Var}\left[\mathbb{E}_{x \sim f}[h_\theta(x)|f]\right] \tag{11}$$
$$\geq \mathrm{Var}\left[\mathbb{E}_{x \sim f}[h_\theta(x)|f]\right] \tag{12}$$

which means the variance of Fiber Monte Carlo lower bounds that of standard Monte Carlo.

There exist many closely related results arising from the study of integral geometry and geometric probability (Santaló, 2004). Crofton's formula (Crofton, 1868) and its measure-theoretic generalization, the Radon transform (Radon, 1917), are similarly concerned with expectations of functions integrated over 'random' lines. These results form a cornerstone of integral geometry, and the theoretical basis for applications in tomographic reconstruction, electron microscopy, and seismology. In the appendix, we provide a proof of correctness using only calculus and basic analysis for the two-dimensional case.

### 3.3 IMPLICIT FUNCTION FORMULATION

We parameterize complex geometries using an implicit function formulation. The interior of a shape is represented using the zero sublevel set[2] of a scalar field $g : \mathbb{R}^d \times \mathbb{R}^m \mapsto \mathbb{R}$ with parameter $\theta \in \mathbb{R}^m$. This is the set $g_z \equiv \{\mathbf{x} \in \mathbb{R}^d \mid g_\theta(\mathbf{x}) \leq 0\}$, with boundary $\mathbf{bd} \, g_z \equiv \{\mathbf{x} \in \mathbb{R}^d \mid g_\theta(\mathbf{x}) = 0\}$. Notice that with this formulation $g$ can be an almost arbitrarily complex scalar field (e.g., a neural network or some general parametric function), which is crucial to parameterizing flexible geometries.

---

[2]Technically, we could use any $\alpha$-sublevel set but we use the zero sublevel set as a canonical form.

Suppose a fiber has nonempty intersection with **bd** $g_z$, and assume that exactly one endpoint of the fiber lies within the interior and the other lies outside. Then we can differentiably compute the point of intersection with the following implicit differentiation framework.

We call $u : \mathbb{R}^m \mapsto (0, 1)$ the 'interpolant' of the fiber; this is an (implicit) function of the parameters with the property that the convex combination $\alpha = \mathbf{x}_s + u(\theta)(\mathbf{x}_e - \mathbf{x}_s)$ lies on the fiber and $g_\theta(\alpha) = 0 \implies g_\theta(\alpha) \in \mathbf{bd} \ g_z$. The optimality condition $F : \mathbb{R}^m \mapsto \mathbb{R}$ that is satisfied when we have determined the correct intersection point between the fiber and the zero-set of the implicit scalar field is:

$$F(u(\theta), \theta) = g_\theta(\alpha) = 0. \tag{13}$$

Given an objective $\mathcal{L} : \mathbb{R}^m \mapsto \mathbb{R}$: the desideratum is the total derivative of $\mathcal{L}$ with respect to the parameters $\theta$. That is,

$$\frac{d\mathcal{L}}{d\theta} = \frac{\partial\mathcal{L}}{\partial u}\frac{du}{d\theta} + \frac{\partial\mathcal{L}}{\partial\theta}. \tag{14}$$

The term $du/d\theta \in \mathbb{R}^m$ can be computed using implicit differentiation as follows. Note that at the solution found by bisection $u^*(\theta)$, we know $F(u^*(\theta), \theta) = 0$ by construction. Then,

$$\frac{d}{d\theta}F(u(\theta)), \theta) = \frac{d}{d\theta}0 \implies \frac{\partial F}{\partial u}\frac{du}{d\theta} = -\frac{\partial F}{\partial\theta} \tag{15}$$

Computing $du/d\theta$ requires only the division of $-\partial F/\partial\theta$ (an $m$-vector) by $\partial F/\partial u$ (a scalar). With this implicit differentiation framework, and computing the other terms using normal automatic differentiation, we are equipped with a differentiable method for computing the points of intersection. From there, we can estimate the line integral using differentiable quadrature.

We assume that $g$ has no more than one zero crossing along each fiber; otherwise, the bisection method does not necessarily converge. Intuitively, if some distribution over scalar fields has no more than one expected zero crossing on some lengthscale bounded by $L > 0$, we would want to inform our choice of fiber length using $L$. In the general case (e.g., neural networks), this bound cannot be derived analytically. We find that it is straightforward to choose short enough fibers in practice, but there still exist pathological functions which have arbitrarily many zero crossings on any positive lengthscale. In the appendix, we discuss a special case using Gaussian processes as a distribution over scalar fields which provides a conceptual framework to reason about the choice of fiber length.

## 4 EXPERIMENTS

### 4.1 IMAGE STYLIZATION

We examine a simplified rasterization-based 2D rendering model (Bangaru et al., 2021) as a case study for using Fiber Monte Carlo as a method for approaching an inverse problem. At the highest level, a rendering engine takes as input a collection of rendering primitives (in this case, 2D shapes with associated color information) and computes an image which depends discontinuously on the spatial configuration of the rendering primitives. We use Fiber Monte Carlo to construct a differentiable renderer which can be used to 'stylize' images by approximately reproducing them from a flexible collection of parametric shapes.

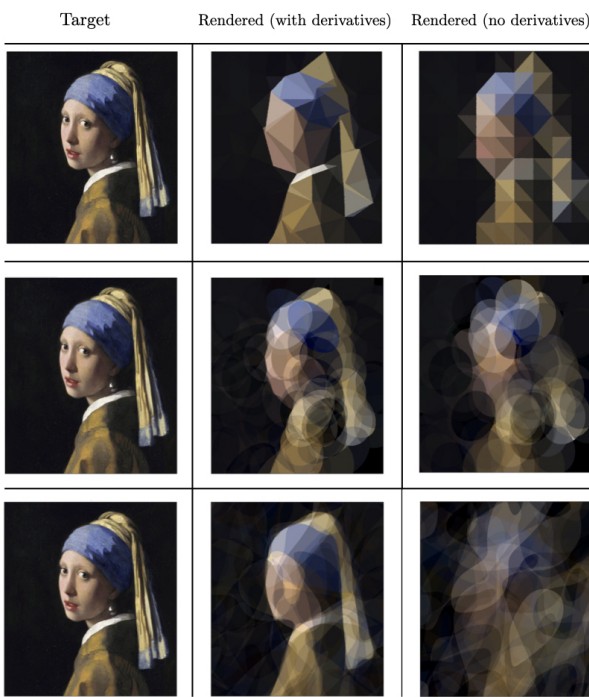

Figure 2: Upper row: image stylization using triangles. Middle row: image stylization with Gaussian density functions as rendering primitives. The zero sublevel sets of a Gaussian density (assuming the output is translated by a negative value) are ellipsoids. Bottom row: image stylization using SIREN neural networks as

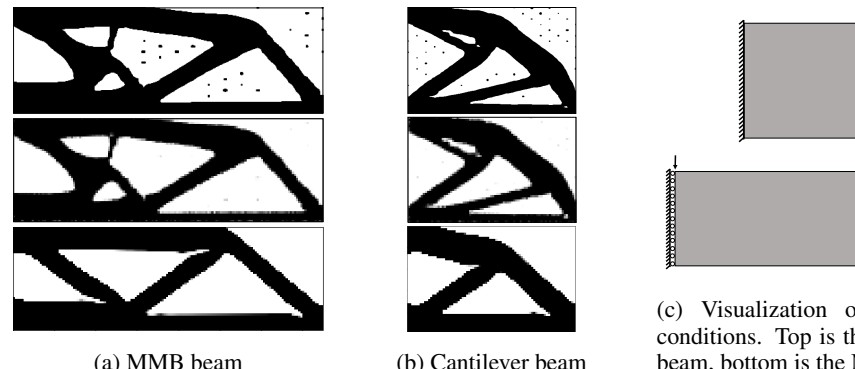

(a) MMB beam  (b) Cantilever beam

(c) Visualization of boundary conditions. Top is the cantilever beam, bottom is the MMB beam.

Figure 3: Designs using topology optimization. Top: Implicit surface learned using fiber sampling. Middle: Pixelized version of implicit surface. Bottom: SIMP Baseline. The qualitative designs of the fiber sampling based algorithm are similar to that of SIMP. One observation with the level-set method is the islands, but these do not contribute to the energy and are not seen when pixelized into the actual structure; they do not particularly represent a disadvantage.

### 4.1.1 PROBLEM DEFINITION

Let $\mathcal{C}_s = [0, 1]^3$ be the set of colors that can be associated with a pixel or rendering primitive. We are given (1) a target image $T \in \mathcal{C}_s^{m \times n}$ (2) a collection of $n_f$ fibers $\mathbf{F} \in \Omega^{n_f \times 2 \times 2}$ (where here the sampling domain $\Omega = [0, 0] \times [m, n]$), and (3) a collection of $k$ 'rendering primitives' $R = \{(\mathbf{r}_i, \mathbf{c}_i)\}_{i=1}^k$, where each rendering primitive is parameterized by a $p$-vector $\mathbf{r}_i$ and is associated with a color $\mathbf{c}_i \in \mathcal{C}_s$.

The forward model (computationally, the rendering engine) maps a collection of rendering primitives, a collection of fibers, and an image size into an image. The rendering primitives are represented by implicit functions $g_i : \mathbb{R}^p \times \mathbb{R}^2 \mapsto \mathbb{R}$ with parameter $\mathbf{r}_i \in \mathbb{R}^p$. That is to say, as in section 3.3, we represent the concrete 'shape' as the zero sublevel set of the implicit function, denoted here with $r_i^z = \{\mathbf{x} \mid g_i(\mathbf{r}_i, \mathbf{x}) \le 0\}$. In exact form, the rendering model corresponds with:

$$f(R, \mathbf{F}, m, n)_{ij} = \sum_{i=1}^k \int_{y=j}^{y=j+1} \int_{x=i}^{x=i+1} c_i \cdot \mathbb{I}((x, y) \in r_i^z). \tag{16}$$

We use Fiber Monte Carlo to estimate this integral, resulting in a function which is differentiable with respect to both the colors and rendering primitive parameters (i.e., the geometry).

We determine the concrete color of a pixel as a convex combination of the color values of the primitives: the weights in the combination are given by the estimated amount of spatial overlap $\mathbf{I}_{ij} \in \mathbb{R}^k$ between each of the $k$ primitives and pixel $(i, j)$, normalized to sum to one.

$$\hat{f}(R, \mathbf{F}, m, n)_{ij} = \mathbf{c}^T \frac{\mathbf{I}_{ij}}{\mathbf{1}^T \mathbf{I}_{ij}} \tag{17}$$

The induced unconstrained optimization problem is: $\text{minimize}_R ||\hat{f}(R, \mathbf{F}, m, n) - T||_2^2$.

### 4.1.2 RESULTS

In fig. 2 we first stylize an image using triangles as rendering primitives, as a direct comparison to previously implemented methods which use smoothing or domain specific languages (Liu et al., 2019; Bangaru et al., 2021). Then we use negated, upward translated, unnormalized Gaussian density functions as implicit functions, resulting in ellipsoidal primitives. Finally, we use neural networks as rendering primitives, to demonstrate the full generality of the implicit function formulation. Each neural network is a sinusoidal representation network (SIREN), introduced in Sitzmann et al. (2020).

## 4.2 Level-Set Topology Optimization

Topology optimization, also known as structural optimization, is an important tool in the design of mechanical systems that has been studied extensively in the past 30 years (van Dijk et al., 2013). The goal is to choose where to place the material of a mechanical structure to minimize a certain objective function (e.g., stiffness of the structure under load), subject to volume constraints. Due to the highly nonconvex and nondifferentiable objectives, various relaxation approaches are employed. The Solid Isotropic Material with Penalisation (SIMP) approach (Andreassen et al., 2011) is the most common, but suffers from various problems such as mesh dependence and nonphysical materials. We show that fiber sampling can be employed as a level-set approach, an alternate topology optimization algorithm. Further details about the two methods are described in the appendix.

Given a regular (e.g., rectangular) domain $\mathcal{D} \subset \mathbb{R}^2$ and a function $g(x, y; \theta) : \mathbb{R}^2 \to \mathbb{R}$ parameterized by $\theta \in \mathbb{R}^m$, we define an *implicit* topology as the set:

$$\mathcal{I} = \{(x, y) \mid (x, y) \in \mathcal{D}, g(x, y; \theta) \leq 0\}. \tag{18}$$

Since during topology optimization the domain of integration changes, it is common to instead compute integrals over the background domain and introduce a Heaviside function.

Solving the PDE underlying the governing physics amounts to finding a function $u : \mathcal{I} \to \mathbb{R}^d$ defined on the implicit domain that minimizes an energy integrated over the domain:

$$u^* = \arg\min_{u \in \mathcal{H}} E(u) = \arg\min_{u \in \mathcal{H}} \int_{\mathcal{D}} H\left[g(x, y; \theta)\right] j\left(\nabla u(x, y)\right) dx dy. \tag{19}$$

where $j(\cdot)$ is a local strain energy derived from material properties.

We discretize the grid and run a linear elasticity simulator with pixel values denoting densities of material at each point. To determine pixel values we rely on the standard approach for level-set topology optimization: for each pixel we measure the area of intersection with the implicit surface. The density of the material is then proportional to the area of intersection. Note that while this will lead to some intermediate densities, they are limited to the boundary of the topology. To make this intersection differentiable with respect to the parameters of the implicit function the usual approach is to smooth the boundary of the implicit surface using a relaxed Heaviside. Instead, we compute this intersection with fiber sampling, which is differentiable even with an exact Heaviside.

We present two industry-standard compliance minimization (stiffness maximization) setups subject to a 50% volume constraint, MMB beam and cantilever, and compare the resulting designs and associated compliance values to a typical SIMP implementation (Andreassen et al., 2011). In compliance minimization, the objective is to minimize the attained minimum strain energy from Eq 19. The two setups differ in terms of boundary conditions, visualized in Figure 3c; the MMB beam represents the right half of a bridge, while the cantilever beam is attached to a wall on the left and pulled down via a force on the bottom right. Qualitative visualizations of the learned topologies are in Figure 3. Quantitative comparisons are below:

|  | SIMP | Fiber Sampling |  | SIMP | Fiber Sampling |
|---|---|---|---|---|---|
| MMB Objective | 0.747 | 0.745 | Cantilever Objective | 0.756 | 0.742 |
| MMB Area | 0.533 | 0.502 | Cantilever Area | 0.533 | 0.503 |

The reported objectives are the unitless ratio of the attained strain energy density to the strain energy of a non-physical 'gray' volume, that is, with 50% material everywhere (lower is better). Although these quantitative results indicate that fiber sampling method potentially offers better performance than SIMP, rigorous comparison of the two methods would require extensive experimentation and this is not the focus of this paper.

## 4.3 Amortized Convex Hulls

We present a case study using Fiber Monte Carlo in an end-to-end optimization problem which amortizes the computation of approximate convex hulls in $\mathcal{O}(nd^2)$ for point clouds comprised of $n$ $d$-dimensional points. Given a point cloud, the output is a set-membership oracle, which essentially answers queries as to whether a point $\mathbf{x} \in \mathbb{R}^d$ lies in the convex hull of the points; these procedures can be tied into fast approximate routines concerning intersections between objects (e.g., as in

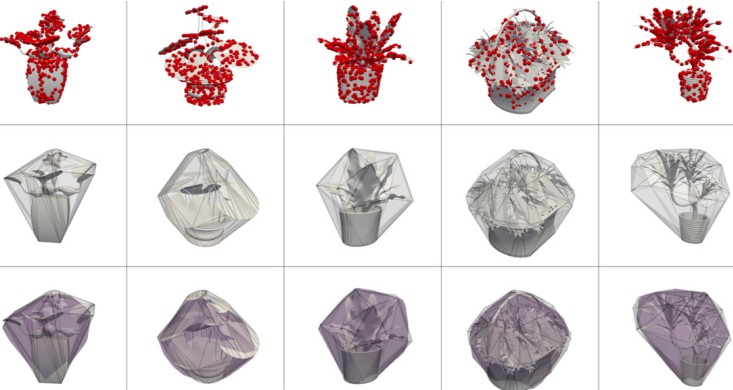

Figure 4: Examples from the (out of sample) validation set. The first row shows the original mesh (not the convex hull) and the point cloud input sampled uniformly from its surface. The second row shows the original mesh and the target convex hull. The last row illustrates the zero set of the predicted implicit function $g_\theta(\mathbf{X})$ in purple.

ray tracing or collisions in computational physics). One interpretation of the oracle is information theoretic: it is an approximate but compressed representation of the convex hull.

### 4.3.1 PROBLEM DEFINITION

We formulate the problem as unconstrained minimization, where the objective is the sample expectation of the negative intersection over union between the shape produced by the forward model (given a collection of $n$ $d$-dimensional points $X$ as input) and the convex hull $\mathbf{conv}X \in \mathcal{X} \subseteq \mathbb{R}^d$.

The forward model $h : \mathbb{R}^m \times \mathbb{R}^{n \times d} \mapsto \mathbb{R}^p$ is a 'hypernetwork' in the sense that it produces a $p$-vector of parameters associated with an implicit function $g : \mathbb{R}^p \times \mathbb{R}^d \mapsto \mathbb{R}$ for points in $d$-dimensional space, this is similar to the modulations approach taken by Dupont et al. (2022). We use an architecture in the spirit of PointNet (Qi et al., 2017), see the appendix for details.

Given the parameters of $\theta$ of the implicit function, $\mathbf{conv}X$, and a collection of $n_f$ fibers $\mathbf{F} \in \mathbb{R}^{n_f \times 2 \times 3}$, we estimate the intersection over union using Fiber Monte Carlo:

$$\mathcal{L} : \mathbb{R}^p \times \mathcal{X} \times \mathbb{R}^{n_f \times 2 \times 3} \mapsto [0, 1] = \hat{\mu}_{\text{FMC}}(g_z(\theta) \cap \mathbf{conv}X) \quad (20)$$

This results in the unconstrained optimization problem: $\min_\theta \mathbb{E}[\mathcal{L}(g_\theta(X), \mathbf{conv}X, \mathbf{F})]$.

### 4.3.2 RESULTS

We evaluate our method on a subset of the ModelNet40 dataset (Wu et al., 2015), which consists of several thousand CAD models partitioned into 40 object categories. We train on approximately 200 point cloud/convex hull pairs drawn from the 'plant' category, sampling 1000 points uniformly from the surface of each plant as input $\mathbf{X} \in \mathbb{R}^{1000 \times 3}$. Evaluation is done using 40 out-of-sample point clouds, also from the plant category.

We sample $m = 40$ halfspace directions at initialization, and then use Fiber Monte Carlo to locally optimize the objective in (19) with respect to the parameters of the hypernetwork. After 200 stochastic gradient updates, we achieve **97.8% accuracy** on the out of sample validation set: several examples are displayed in fig. 4. The target hulls have in general more points than we use halfspaces; this is the sense in which our set membership oracles offer a compressed representation of $\mathbf{conv}X$.

## 5 CONCLUSION

Fiber Monte Carlo enables gradient-based optimization of objectives which contain integrals with parametric discontinuities. Implemented within a standard automatic differentiation framework with utilities for implicit differentiation, the technique can be applied widely to problems in physical simulation and design, topology optimization, computational geometry, and graphics. Cast as a conditional Monte Carlo estimator, FMC inherits the analytic foundation of Monte Carlo methods from physics and statistics, which accelerates the work necessary to interrogate the theoretical character of the approach and understand its implications as a generic statistical estimator.

Interacting meaningfully with discontinuities in any gradient method requires compromises and tradeoffs. Importantly, our technique is likely to scale poorly with the dimension of the integration

domain. The geometry of high-dimensional spaces implies that via concentration phenomena (e.g., sphere hardening (MacKay et al., 2003)) volumes tend to be confined to regions of smaller relative extent. In the absence of more sophisticated bounding regions to sample from, naively one would expect that achieving a constant variance estimate requires a number of fibers that scales exponentially with dimension.

The work we present here arose as a solution to real problems we encountered in projects interacting with generative design of physical systems and differentiable simulation. With that in mind, one future thread of work is integrating the contributions here to problems at the intersection of computational modeling and simulation and generative design. Along a more theoretical axis, we believe Fiber Monte Carlo can also be employed as a general-purpose methodology for uniformly sampling from a large class of manifold structures, an open problem spanning computational geometry, numerical partial differential equation (PDE) solving, and graphics. Crisper theoretical analysis and experimentation would be required to adjudicate the veracity of that hypothesis.

## ACKNOWLEDGEMENTS

The authors would like to thank all members of the Princeton Laboratory for Intelligent Probabilistic Systems for valuable feedback. This work was supported by NSF grants IIS-2007278 and OAC-2118201. JCB was supported in part by a 2022 Caltech Rypisi SURF Fellowship.

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

# A APPENDIX

## A.1 NOTATION

| | |
|---|---|
| **conv** $C$ | The *convex hull* of a finite set $C$. |
| **bd** $X$ | Boundary set of $X$, the set difference of the closure of $X$ with its interior. |
| **dist** $(\mathbf{x}, C)$ | The (Euclidean) distance between a point x and a set C, i.e., $\inf\{||\mathbf{x} - \mathbf{y}||_2 \mid \mathbf{y} \in C\}$. |
| **len** $f$ | Length of a fiber $f$, which is the norm of the difference of its endpoints $||\mathbf{x}_s - \mathbf{x}_e||_2$ and generally a fixed positive quantity $\ell > 0$. |

## A.2 PROOFS

In this section we provide a (simple) proof which shows correctness of the Fiber Monte Carlo estimator (and its derivatives) in the two-dimensional case. We use the following lemma in our proof of correctness:

**Lemma 1**

$$\int_{y=-1-\ell}^{y=1} \int_{t=y}^{t=y+\ell} \mathbb{I}[t \in [-1, 1]] f(t) dt \, dy = \ell \int_{t=-1}^{t=1} f(t) dt. \tag{21}$$

The justification is:

$$\int_{y=-1-\ell}^{y=1} \int_{t=y}^{t=y+\ell} \mathbb{I}[t \in [-1, 1]] f(t) dt \, dy = \int_{t=-1}^{t=1} \int_{y=t}^{y=t+\ell} f(t) dy \, dt$$

$$= \ell \int_{t=-1}^{t=1} f(t) dt.$$

The first equality holds if $f$ is absolutely integrable (i.e., interchanging of integrals/limits is valid), and the fact that $t$ is restricted to the closed set $[-1, 1]$ via the indicator function. The second step follows since the resultant integrand has no dependence on the innermost variable of integration, so the contribution is simply a constant value $\ell$.

**Correctness**
For simplicity consider an absolutely integrable function $g_\theta : \mathbb{R}^2 \to \mathbb{R}$ supported in the square domain $[-1, -1] \times [1, 1]$, and say we sample a fiber of length $\ell > 0$ with vertical direction. We will prove that the line integral of $g_\theta$ over the fiber is an unbiased estimate of

$$I(\theta) = \int_{x=-1}^{x=1} \int_{y=-1}^{y=1} g_\theta(x, y) dy \, dx.$$

Concretely, the fiber has endpoints $(x_0, y_0)$ and $(x_0, y_0 + \ell)$, where $x_0 \sim \text{Unif}[-1, 1]$ and $y_0 \sim \text{Unif}[-1 - \ell, 1]$. The line integral over the fiber is then:

$$\hat{I}_1 = \frac{1}{\ell} \int_{t=y_0}^{t=y_0+\ell} \mathbb{I}[t \in [-1, 1]] g_\theta(x_0, t) dt.$$

The expectation of this estimate over $(x_0, y_0)$ is:

$$\mathbb{E}[\hat{I}_1] = \frac{1}{\ell} \int_{x=-1}^{x=1} \int_{y=-1-\ell}^{y=1} \int_{t=y}^{t=y+\ell} \mathbb{I}[t \in [-1, 1]] \, g_\theta(x, t) \, dt \, dy \, dx \tag{22}$$

$$= \frac{1}{\ell} \int_{x=-1}^{x=1} \ell \int_{t=-1}^{t=1} g_\theta(x, t) \, dt \, dx \tag{23}$$

$$= \int_{x=-1}^{x=1} \int_{y=-1}^{y=1} g_\theta(x, y) \, dy \, dx \tag{24}$$

$$= I(\theta), \tag{25}$$

where equation 23 follows from Lemma 1, and equation 24 simply substitutes $y$ for $t$ to correspond with the original quantity. This proves correctness of the estimator.

**Correctness of Derivatives**

Correctness of the Fiber Monte Carlo derivative estimate follows straightforwardly from Lemma 1. First note that we cannot simply interchange differentiation and integration in the quantity

$$\frac{\partial}{\partial \theta} \int_{y=-1}^{y=1} f_\theta(y) dy, \tag{26}$$

since $f_\theta$ contains a discontinuity with respect to $\theta$ by assumption. Put another way, interchanging differentiation and integration is valid when interchanging integrals and limits is valid; a condition to establish this validity is that the integrand is continuously differentiable in $\theta$. Despite the fact that we cannot straightforwardly interchange integration and differentiation here, we know the following equality holds from Lemma 1:

$$\frac{\partial}{\partial \theta} \int_{y=-1}^{y=1} f_\theta(y) dy = \frac{\partial}{\partial \theta} \frac{1}{\ell} \int_{t=y}^{t=y+\ell} \mathbb{I}[t \in [-1,1]] f_\theta(t). \tag{27}$$

The integral on the right hand side is almost everywhere differentiable. Assuming we can analytically compute the integral (e.g., via quadrature), we can use automatic differentiation to compute derivatives. Given Lemma 1, we can write derivatives of the estimator as,

$$\frac{\partial}{\partial \theta} \hat{I}(\theta) = \frac{1}{\ell} \frac{\partial}{\partial \theta} \left( \int_{t=y}^{t=y+\ell} \mathbb{I}[t \in [-1,1]] f_\theta(t) dt \right) dy. \tag{28}$$

## A.3 Technical Details

### A.3.1 Initialization of Scalar Fields

In applications like image stylization, we explicitly parameterized the scalar fields corresponding to rendering primitives so that at any initialization of the parameters, there were fibers intersecting the zero sublevel set of the field. Otherwise, there would be no gradient signal using this parameterization. In practice so long as the scalar field has mean value equal to zero and some zero crossings in the sampling domain, one can push useful derivatives through it during optimization.

### A.3.2 Zero-Crossings and Implicit Differentiation

Our implicit differentiation formulation operates under the tacit assumption that the function does not vary 'too much' on the length-scale of any given fiber. This is required to guarantee convergence of the bisection method in computing the intersection point of a fiber whose endpoints evaluate to different signs: we need to ensure the function has no more than one zero-crossing along the fiber.

In practice, this condition is not problematic: in many applications of interest the function is piecewise linear. Moreover, even if it were not, the length of the fibers is a degree of freedom which can be chosen to avoid the circumstance of multiple zero-crossings.

In this section we explore a pedagogical case, using functions sampled from Gaussian processes as a surrogate: not meant as a practical field guide but a conceptual picture of what can go wrong if the fiber length is too long (which we make precise shortly).

We consider a Gaussian process with mean function $m$ and kernel $k$ over real scalar field functions $g(x) \sim \mathcal{GP}(m(x), k(x, x'))$. Assume a zero mean Gaussian process with unit marginals and standard Gaussian kernel:

$$k(x, x') = \exp(-(1/(2L^2))||x - x'||_2^2).$$

Then it follows that the expected number of zeros over a fiber of length $\ell$ is $\mathbb{E}N = \frac{\ell}{\pi L}$ (Ylvisaker, 1965). This result dates back to (Rice, 1944) who derives it from a different (but equivalent) set of assumptions. Suppose we want to bound the probability of more than one zero crossings over $\ell$ by 0.01, then we choose $\ell = (2\pi L)(.01)$ by Markov's inequality. In this simple case, the interpretation of the bound is: provided the fiber length to is less than 6% of a given characteristic lengthscale of variation associated with the functions, on average fewer than 1% of the fibers we sample result in the bisection method failing to converge.

In the general case, we of course lack an explicit model for $g$. The point of the argument is to certify the intuitive idea that the fiber length should be chosen strictly smaller than some characteristic 'lengthscale of variation' of the function, we use the simplified case in order to make this notion precise.

There is a straightforward analogy to simple Monte Carlo. In simple Monte Carlo, the variance of the estimator is proportional to the variance of the estimand: analogously, the more 'variable' the implicit function, the shorter (and thus, more) fibers are needed to achieve the same estimate variance (all else equal).

### A.4 EXPERIMENTAL DETAILS

#### A.4.1 IMAGE STYLIZATION

We used a single Nvidia RTX 3080Ti GPU for differentiable rendering, with the wall clock time to optimize against a single image ranging from 5-15 minutes, depending on the number of rendering primitives used. Rendering with triangles, we used 75 triangles, initialized by sampling their vertices uniformly at random from the space spanned by the image, and then computing the triangles with the Delauney triangulation of the vertices (during optimization we disallowed updates that would cause any vertices to migrate outside of the image). Rendering with negated, shifted, Gaussian densities, we used 50 implicit functions, whose means were sampled uniformly over the image space, with precisions sampled uniformly from a fixed range (determined visually so that the primitives were large enough at initialization). Rendering with SIREN networks, we used 20 networks with a frequency $\omega = 10$. Each network had 3 hidden layers, each of size 8. For all rendering experiments we optimized using Adam with a step size of .001.

#### A.4.2 TOPOLOGY OPTIMIZATION

The most common relaxation pixelizes the domain, relaxes the problem to contain "intermediate" amounts of material in each pixel, and runs an optimization algorithm over the relaxed densities of each pixel to find the optimal topology. To ensure physicality, the method penalizes intermediate values. This is called SIMP (Andreassen et al., 2011), and although by far the most common approach, can often lead to mesh dependence ("checkerboarding") and highly nonphysical materials. To counteract this, various filtering and regularization techniques have been designed to stabilize the resulting designs. Despite these limitations, SIMP-type approaches are widely considered the industry standard in topology optimization.

An attractive alternative approach to topology optimization involves representing the domain as a level-set of a higher dimensional function. This approach has garnered much research (van Dijk et al., 2013) as it promises better continuity and mesh independence, but a problem it has is that it involves differentiating through integrals with parametric discontinuities (usually used to compute potential energy over the domain defined by a level-set). The common way to compute this involves using a smoothed Heaviside-function in the integral, but with fiber sampling we can do it exactly.

Although with the level-set approach any type of implicit surface is in theory possible (e.g., neural implicit fields), to demonstrate fiber sampling and avoid additional difficulties of using neural fields for topology optimization we stick to the most common parameterization: a grid of localized basis functions forming a piecewise linear function over the background domain, as in Belytschko et al. (2003).

### A.4.3 AMORTIZED CONVEX HULLS

The hypernetwork produces a pointwise embedding for each input point (using a single MLP), aggregates those embeddings dimension-wise with a symmetric function (e.g., maximum, sum, average), and then produces the implicit parameters as output by transforming this global embedding (using another MLP).

For the ModelNet40 experiment we sampled 40 normalized halfspaces whose direction was chosen uniformly over the unit cube. We preprocessed the data by translating the mean of the point clouds to the origin, and scaling so that the point cloud fit within the unit 2-norm ball, each point cloud was comprised of 1000 samples drawn uniformly over the surface of the original plant mesh. The hypernetwork used a pointwise MLP with three hidden layers of size 200, dimension-wise maximum as the symmetric aggregation, and an output MLP with three hidden layers of size 200. We trained on a single Nvidia RTX 3080Ti GPU using Adam with step size .001: training took 2 hours wall clock time.

### A.4.4 REPRODUCING RESULTS

We will release our generic utilities via a public Python package. In the effort of minimizing the difficulty of reproducing the results detailed in this work, we will also publish an associated Dockerfile that can be used to build a cross-platform container image to reproduce any of the figures displayed in this paper.

