# OpenReview forum: "Fiber Monte Carlo"
_ICLR.cc/2024/Conference — ICLR 2024 poster_

### Official Review · Reviewer_qHwo · 2023-10-30

**Soundness:** 3 good
**Presentation:** 3 good
**Contribution:** 2 fair
**Rating:** 6
**Confidence:** 3

**Summary:**

Proposes a new formulation of Monte Carlo Integration/Estimation that samples line segments on the integration domain as opposed to points.  This is of particular relevance in gradient-based optimization for inverse problems, where discontinuities in gradients of the MC estimate with respect to input parameters are either non-differentiable, or lead to incorrect results.  By sampling line segments instead of points, the parametric gradients become well-defined and differentiable.

**Strengths:**

The paper is generally well-written, and addresses an established problem space that is relevant to a broad range of applications as enumerated by the authors.  The central idea itself (sampling "fibers" rather than points) is interesting, and claims to be a general formulation that avoids domain-specific solutions and can be applied within a generic auto-differentiation framework.  The theoretical correctness of their proposed sampling method is proved (i.e. that it remains an unbiased estimator).

**Weaknesses:**

While the broad application areas presented are impactful, the actual evaluation is very limited.  Specifically, the rendering application is evaluated using only qualitative (not quantitative) results, on a single scenario.  The topology optimization example is also evaluated on only a single example -- while it is understandable that the authors specifically acknowledge that in-depth evaluation of this task is lacking, stating that this is not the central focus of the paper, it brings into question the meaningfulness of this experimental result.  In the case of the convex hull example, no comparison to alternative methods is provided.  (I would personally find a slightly more in-depth investigation of fewer applications to be more enlightening, though I do not expect the authors to drastically alter their existing experiments).

There seem to be missing key references w/r/t the claimed applications, for example, in the case of differentiable rendering and simulation.  Are such references considered to be out of the scope of the paper?  As mere examples, in the case if differentiable rendering (by no means an exhaustive list):
- Merlin Nimier-David and Delio Vicini and Tizian Zeltner and Wenzel Jakob.  Mitsuba 2: A Retargetable Forward and Inverse Renderer.
- Tzu-Mao Li, Miika Aittala, Frédo Durand, Jaakko Lehtinen.  Differentiable Monte Carlo Ray Tracing through Edge Sampling.

**Questions:**

- In some applications (e.g. Monte Carlo Path Tracing Rendering), the integration domain must be sampled according a non-uniform probability distribution.  This is commonly achieved by inverse-transform sampling, whereby random variables are sampled from a uniform distribution, then mapped to the target distribution via its inverse-CDF.  Is Fiber Monte Carlo compatible with such sampling requirements?

---

> ### Author Response · Authors · 2023-11-15
>
> Thank you for taking a close read of our work and dedicating the time to review it. We agree that a comprehensive and deep investigation of just one or two applications is often the right approach. We felt some tension around this ourselves, but in this work, our aim was to establish the generality of our method, as applied to several (seemingly) disparate areas of application.
>
> The tradeoff in doing so is that each of these application areas are research areas in their own right of course (which we can’t do justice to within a few pages of writing). That is all to say, the core (application-oriented) claim we are making is that our method solves a novel problem with a surprisingly far reach in applications. Further work could, for example, exhaustively analyze fiber Monte Carlo applied to topology optimization, but this work is about explicating the fundamental method. Note that we evaluate topology optimization on two examples, not one (though again, we of course could have evaluated three, or seven).
>
> We considered differentiable rendering more broadly to be out of scope of the paper, exactly in the spirit written above (although we are fans of the work you listed). We chose to consider image stylization specifically because it is an appropriate target for fiber Monte Carlo, and no general method had been used to achieve stylization in the literature. Further, it’s worth noting that image stylization is an application which lacks a great/insightful quantitative metric. Technically one frames the application as an optimization problem, but this isn’t the complete picture since solving the problem as posed would not actually result in a stylized image (the minimum of the objective is attained by rendering the original target image perfectly, which hardly “stylizes” it). Here we felt the relevant comparison to our method is attempting to stylize the image without using a procedure to estimate derivatives with respect to geometry (instead only with respect to color), and we provide the results attained by doing so in a side-by-side comparison (see figure 2).
>
> Finally, our application of approximately computing convex hulls is not an application that has been seriously examined in the literature, to our knowledge. We display out-of-sample examples from our data (figure 4) and report out-of-sample accuracy on the intersection over union objective. We could devise yet another method to approach the problem, and then attempt to compare two novel methods, but we don’t feel that would helpfully contextualize the results given this problem has not been approached previously in the literature to our knowledge.
>
> You ask a great question about non-uniform sampling. We have not explicitly attempted to work out the details of the scenario you mention. Speculatively, if one can push the fibers through the inverse function (possibly representing the fibers as parametric curves), it should still be possible to apply our implicit function formulation to estimate the integrals. This is an interesting use-case that should probably be the focus of a follow-up study. Happy to hear your thoughts or speculations on this as well!

---

> > ### Comment · Reviewer_qHwo · 2023-11-22
> >
> > Thanks to the authors for your responses.
> >
> > I remain convinced that this is an interesting idea.  My main concern is that the generality of the proposed method is still not overwhelmingly clear.  For example, differentiable rendering (i.e. MC path tracing) is considered to be out of the scope of the paper (which is not inherently a problem), but it brings into question whether this method would actually generalize to such real, well-established applications for MC integration.
> >
> > Specifically regarding the question of inverse-transform sampling, in my opinion this would be a very valuable section to include in the paper, as sampling non-uniformly over a domain surely opens the doors to additional applications, and would further make the case for the generality of the proposed method.
> >
> > I will maintain my original score.

---

### Official Review · Reviewer_sad6 · 2023-11-01

**Soundness:** 4 excellent
**Presentation:** 4 excellent
**Contribution:** 4 excellent
**Rating:** 8
**Confidence:** 4

**Summary:**

The paper considers a novel differentiable integral estimator that operates by sampling lines instead of points. The method generalizes to a relatively broad swath of low-dimensional integrals with discontinuous integrands, allowing for quick prototyping and broad application to a variety of scenarios, as shown in the experiments.

**Strengths:**

I have reviewed this paper previously as a NeurIPS submission, and it has changed relatively little. That is to say, I was pretty positive on it back then, and my opinions remain unchanged.

* The problem considered is an important and wide-ranging one. The method is implemented in a common library, so should allow for ease of use by a variety of practitioners in other application domains.
* I still think the presentation is quite clear. A previous reviewer challenged it as imprecise, but I do not share their opinion and felt they were nitpicking.
* I appreciated the validation across three relatively different scenarios.

**Weaknesses:**

* The restriction to low-dimensional integrands may cut out some desired use cases. This is more of an inherent challenge perhaps that any Monte Carlo-based method would suffer.
* As before, I think it would be interesting to see the method compared to bespoke algorithms in those particular use cases. I expect, of course, that the given method will do worse, but if it is comparable that would be a win, given its much broader applicability.

**Questions:**

1. Can you delineate what has been changed from the NeurIPS submission?
2. In the previous NeurIPS discussion, you referenced a plot of variance as dimension increased. Can you include this again in supplementary? I think it's useful for noting this limitation.

---

> ### Author Response · Authors · 2023-11-15
>
> Dear reviewer, great to hear from you again! Thanks for taking the time to review (again) our work. We’ll be brief in our response to your two questions.
>
> 1. As you mentioned, one previous reviewer was concerned about the level of precision offered in our technical arguments. In this revision, we treat an argument for correctness using measure theory directly and cite relevant/similar results from integral geometry. We felt concerned that this argument might not be as generous/clear as our previous one, so we also cleaned-up and simplified an argument for correctness using just calculus, which is in the appendix. The writing overall was picked through and re-written throughout to improve clarity, and we feel the paper reads much better this time around.
> 2. Absolutely, we will include this plot: thanks for this suggestion.

---

> > ### Comment · Reviewer_sad6 · 2023-11-22
> > **Response acknowledged**
> >
> > Thank you for the response here. I still feel pretty positive about the work, and the tack that it is taking (as a more general tool, as opposed to one aiming to beat tailored, state-of-the-art methods). I'll leave my score unchanged, but look forward to a robust discussion between reviewers to reach a consensus.

---

### Official Review · Reviewer_mMv5 · 2023-11-03

**Soundness:** 2 fair
**Presentation:** 3 good
**Contribution:** 3 good
**Rating:** 6
**Confidence:** 4

**Summary:**

This paper proposes a way to estimate derivatives of integrals of discontinuous functions with respect to parameters. Sampling line segments rather than points produces an empirical estimate that is continuous and (almost-everywhere) differentiable. Assuming the discontinuous integrand is parametrized by a superposition of indicator functions parametrized by implicit level sets, the derivatives of integrals over individual line segments can be calculated by differentiating the implicit functions. The method is demonstrated on several problems: 2D inverse rendering, topology optimization, and convex hull approximation.

**Strengths:**

I like the simplicity and elegance of the central idea: sample extended geometric objects (in this case line segments) rather than points, and use the geometry to get better estimates. The authors point out the long history of this idea in mathematics, dating back at least to the Crofton formula, Radon transform, and integral geometry. These are ideas that I think have yet to be fully exploited in neural rendering/neural fields/inverse graphics. One related work I would suggest citing is the following, which also involves the evaluation of neural fields along lines rather than points:

- V. Sitzmann, S. Rezchikov, B. Freeman, J. Tenenbaum, and F. Durand, “Light Field Networks: Neural Scene Representations with Single-Evaluation Rendering,” in Advances in Neural Information Processing Systems, Curran Associates, Inc., 2021, pp. 19313–19325.

**Weaknesses:**

The applications seem less than convincing to me. It seems clear that the image stylization is meant to stand in for inverse graphics more generally, but it is unclear to me what advantage the type of representation to which this method is wed (superposition of sublevel sets) would be advantageous over a continuous field representation, which would obviate the need for the fiber sampler.

The application to topology optimization is more compelling in that topology optimization generally seeks solutions in indicator functions. However, the choice to represent the indicator function on a grid seems puzzling when a great advantage of a sampling-based estimator coupled to automatic differentiation is the flexibility of the underlying representation.

The convex hull application seems the weakest to me. It looks like the generated approximate hulls are not even convex, nor do they contain all the points, so it is unclear how they would be useful in downstream applications such as collision detection. In any case, classical convex hull algorithms in low dimension are plenty fast and come with correctness and performance guarantees. Accordingly, any proposal to replace them with a learned "oracle" must clear a very high bar.

More generally, only a few results are shown, and the numerical comparisons in the topology optimization case only show a marginal improvement in objective function. It would be good to include at least a few more test cases and comparisons.

**Questions:**

- It seems like the method is limited to settings where the integrand is described by a superposition of indicator functions parametrized by implicit sublevel sets. But if one is going to use such a representation, why not just use a continuous implicit function directly. For example, rather than using a superposition of level sets of SIREN networks, one could just use SIREN directly. In what application domains do you see this sort of representation being uniquely advantageous?
- The authors repeatedly emphasize that the method is only applicable in low dimension. Why is this? What would it take to extend the method to general dimension? What about to spaces other than $\mathbb{R}^n$?
- What are the island artifacts in the topology-optimized solutions using fiber sampling? Why do they appear?
- What would it take to extend the method to more general geometry representations for topology optimization rather than just grid discretization?

---

> ### Author Response · Authors · 2023-11-15
>
> Thank you for your helpful feedback. We sincerely appreciate the time and effort you’ve taken to read the work closely and provide valuable feedback. As we understand it, you’re chiefly concerned about the experimental evaluations. We’ll try to provide some context around the experiments and address your specific questions.
>
> At a broad level, we feel that the contribution of this work is a novel method for solving problems which were not previously generically solvable. We do not take the position that our performance on any of the applications surveyed is superior to state of the art. Our position is that performance is clearly comparable, even without domain-specific adaptations to the core method.
>
> All that said, we address your specific concerns in the following paragraphs.
>
> **********************************Image Stylization**********************************
>
> You write that “it seems clear that the image stylization is meant to stand in for inverse graphics more generally”. As follows from the paragraph above: we do not mean to imply or claim that the image stylization application we present is meant to “stand in for inverse graphics more generally.” We chose image stylization in particular (as a specific sub-problem within inverse graphics) because the objective function contains an integral with a parametric discontinuity; this makes it an appropriate target for fiber Monte Carlo.
>
> **************************Topology Optimization**************************
>
> It is true that the geometry representation is still a grid. In general, level-set topology optimization is known to be very difficult to do stably. Using flexible representations, although in theory possible, is difficult to get right. Existing literature has focused on grid representations, so to align with existing work we chose to stick with the grid representation. Also of note is that topology optimization is a large field with a vast literature, and we do not expect to show that our method is the “best” out of the box on two examples; the marginal improvement in objective function is possibly only incidental. A rigorous evaluation of fiber sampling with topology optimization potentially deserves its own paper; here we simply present that it performs reasonably out of the box compared to a standard approach from the literature, since the focus of this work is about explicating the fundamental method of fiber sampling.
>
> ************************************************Approximate Convex Hulls************************************************
>
> Visually, we see that in certain portions of the image it is not immediately apparent that the approximate hulls are convex sets (the pale coloring makes it hard to distinguish); this is useful feedback for regenerating finalized figures with a better color set. Rest assured that this is a plotting artifact: the approximate hulls are convex by construction. We take the intersection of a finite set of half-spaces. Since a halfspace is a convex set, and the intersection of a finite set of convex sets is itself convex, the approximate hulls are convex (for any value of the parameter).
>
> It is true that our approximate hulls are not supersets of the true convex hulls (i.e., they do not contain all the points, as you mention). This why we use the term “approximate” convex hulls (necessarily convex sets which are not necessarily equal to the convex hull). We do not propose the replacement of classical convex hull algorithms, and completely agree that such a proposal would require significantly more comprehensive analysis (in fact, this is an immense area of study and could be the subject of an entire paper).

---

> > ### Author Response · Authors · 2023-11-15
> >
> > ******Responses to Questions******
> >
> > 1. There may be some confusion here as to the notion of a “primitive” in the image stylization experiment. The core property of a primitive is its associated color. The single sentence answer to your question is: if we had used any one implicit function (i.e., primitive): we would have precisely one color in the image. Put another way, you can think of each individual SIREN network for $i = 1, \dots n$ as having associated with it a particular color, which allows us to render non-monochromatic images. When $i=1$ we have just $c_1$ and thus we are optimizing the spatial distribution of the color $c_1$ across the image canvas. It is not the case that the method is limited to settings where the integrand is described by a superposition of indicator functions parameterized by implicit sub-level sets, although this is absolutely a useful formulation to flexibility represent geometry. As a counterexample, consider our first stylization with triangles: which applies the method to an (1) an integrand which does not contain any implicit functions and (2) does not parameterize geometry with sub-level sets. That said, the implicit function formulation is hardly a limitation: in fact it is the key to parameterizing nearly arbitrary geometries.
> > 2.  The method is limited to low/small dimension since it estimates an integral over the sampling domain. In the general case, numerically computing an integral is a hard problem when the dimension is large. It is difficult to be precise about this in brief, but roughly speaking the difficulty of the computation depends exponentially on the dimension. But suppose, for sake of argument, that you could compute the integral. Then we would expect that the variance of the derivative estimates would increase with dimension. However, this should not be confused with a requirement that the *********parameter********* be small in dimension. As we write in the paper: “it is important to note that Fiber Monte Carlo is appropriate provided the domain of integration has small dimension. This is unrelated to the dimension of the parameter: we show that there is no difficulty in training large parametric models (e.g., neural networks) with this frame-work.” Your question about extension beyond $\mathbb{R}^n$ is quite interesting. Which spaces do you have in mind?
> > 3. The island artifacts are common in level-set topology optimization, and do not lead to degradation in quality. They appear simply because a disconnected component receives no gradient signal in optimization. They can be removed in post-processing via a connected components algorithm, but we left them in to present the exact output of our method.
> > 4. See the main “Topology Optimization” section of our response.
> >
> > We hope this addresses your concerns, but we’re happy to follow up on any aspect in more detail or respond to further questions.

---

> ### Comment · Reviewer_mMv5 · 2023-11-23
> **Response to author rebuttal**
>
> **Image stylization:** I accept that this is an application in its own right, and it is not intended to stand in for inverse graphics more generally.
>
> **Approximate convex hulls:** The authors say that the approximate convex hulls are formed as the intersections of half-spaces, which would indeed make them convex by construction. However, the fact that the hypernetwork predicts intersections of half-spaces is only alluded to in both the paper and the supplementary material. It would be good to make this explicit. The text only specifies that "[t]he forward model $h : \mathbb{R}^m \times \mathbb{R}^{n\times d} \to \mathbb{R}^p$ is a ‘hypernetwork’ in the sense that it produces a $p$-vector of parameters associated with an implicit function." Later it is mentioned that half-spaces are sampled, but it is unclear from the text how they are used.
>
> Moreover, it is still unclear to me where an approximate convex hull algorithm can be useful if it is not guaranteed to be conservative. E.g., in collision detection, false positives are OK as they can later be resolved, but false negatives are not.
>
> **Topology Optimization:** This seems like the most compelling application, and it would be great to expand on this a bit more, perhaps by cutting or reworking the approximate convex hull section.

---

### Meta-Review · Area_Chair_xL9h · 2023-12-08

**Metareview:**

The paper tackles the problem of integrating discontinuous integrands, common in areas like topology optimization and graphics, into machine learning. Traditional Monte Carlo methods, used for these integrals, don't work well with automatic differentiation, a key tool in machine learning. This issue limits the application of machine learning in fields with such discontinuities. The authors propose a new approach that modifies the Monte Carlo method by sampling line segments instead of points, making it compatible with automatic differentiation. They demonstrate its effectiveness in various applications like image stylization and computational geometry.

**Justification For Why Not Higher Score:**

While it's understood that no method excels in every scenario, the proposed method has a somewhat limited range of applications, likely appealing to a smaller subset of the community. However, I anticipate that the underlying ideas will be adopted and modified to address other related challenges in broader contexts.

**Justification For Why Not Lower Score:**

The text effectively empirically demonstrates the method's value across several scenarios, making a strong argument for its publication.

---

### Decision · Program_Chairs · 2024-01-16

Accept (poster)